# Photometric Long-Range Positioning of LED Targets for Cooperative Navigation in UAVs

**Laurent Jospin** [1] , **Alexis Stoven-Dubois** [2] **and Davide Antonio Cucci** [1,*]

1    Geodetic Engineering Laboratory, École polytechnique fédérale de Lausanne, 1015 Lausanne, Switzerland
2    Mobility Department, Vedecom, F-78000 Versailles, France
*    Correspondence: davide.cucci@epfl.ch; Tel.: +41-2169-33292

**Abstract:**   Autonomous flight with unmanned aerial vehicles (UAVs) nowadays depends on the availability and reliability of Global Navigation Satellites Systems (GNSS). In cluttered outdoor scenarios, such as narrow gorges, or near tall artificial structures, such as bridges or dams, reduced sky visibility and multipath effects compromise the quality and the trustworthiness of the GNSS position fixes, making autonomous, or even manual, flight difficult and dangerous. To overcome this problem, cooperative navigation has been proposed: a second UAV flies away from any occluding objects and in line of sight from the first and provides the latter with positioning information, removing the need for full and reliable GNSS coverage in the area of interest. In this work we use high-power light-emitting diodes (LEDs) to signalize the second drone and we present a computer vision pipeline that allows to track the second drone in real-time from a distance up to 100 m and to compute its relative position with decimeter accuracy. This is based on an extension to the classical iterative algorithm for the Perspective-n-Points problem in which the photometric error is minimized according to a image formation model. This extension allow to substantially increase the accuracy of point-feature measurements in image space (up to 0.05 pixels), which directly translates into higher positioning accuracy with respect to conventional methods.

**Keywords:**  cooperative navigation; relative positioning; image formation model; photometric error; PnP

## 1. Background and Motivations

Unmanned aerial vehicles (UAVs) are a well-established tool for surveyors and engineers, as they can provide cost-effective, high resolution, aerial imagery for mapping and 3D reconstruction of natural and artificial structures [1]. One important limit of the current UAV technology is the dependency on the Global Navigation Satellite Systems (GNSS) coverage as positioning information is generally required for guidance. Good reception of the GNSS signals is also required for cm-level mapping without the need to survey several ground control points [2,3]. The dependency on the GNSS reception severely limits the applicability of UAVs each time the sky is in large part occluded: the geometry of the GNSS constellation degrades and the position fix becomes irregular, inaccurate (due to multipath effect) and unreliable. This is very common in mountainous environment, e.g., see Figure 1, or near large natural or artificial structures, such as bridges, dams, in urban canyons, etc. In such conditions, autonomous flight is risky and most commercial platforms prevent the take-off.

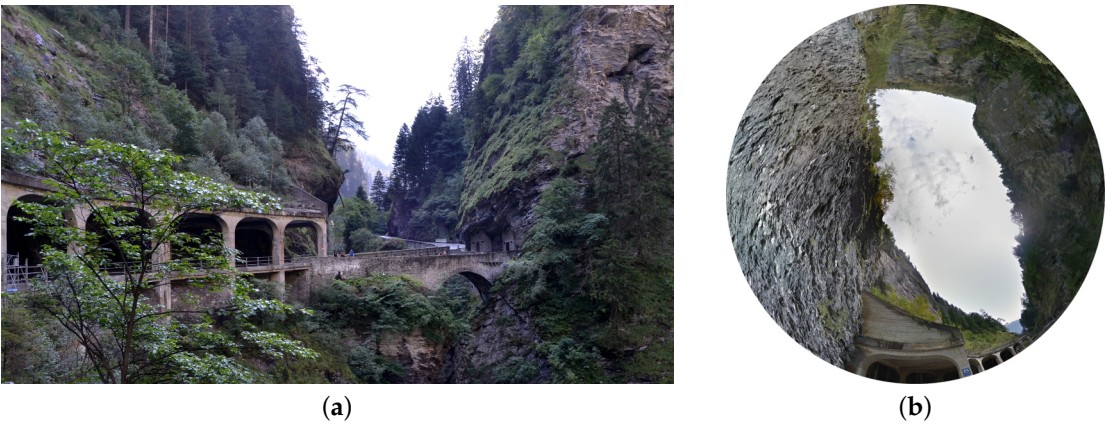

**Figure 1.** A rock fall protection gallery in a narrow gorge (46.6627722, 9.4484218). Approximately 80% of the sky is occluded by natural or artificial structures and vegetation. (**a**) Via Mala, Graubünden (CH). (**b**) Sky plot.

Many researchers are currently focusing on developing navigation solutions for GNSS denied environments. Most of these are vision-based or involve some sort of multi-sensor fusion (visual-inertial navigation being the most popular [4–6]). These methods rely on exteroceptive sensors and thus are inherently dependent on the environment (e.g., enough texture has to be present in the scene) and it is difficult to certify or guarantee their performances. This is the reason why many excellent research prototypes exist whereas no commercial product is currently shipped with full fledged visual-inertial navigation.

If the environment can be freely structured, e.g., by placing landmarks, beacons, or other sensors, positioning information may be obtained similar to the one provided by GNSS. Once such landmarks have been identified, the navigation problem can be solved, e.g., as presented in works by the authors of [7,8]. Another solution is given by ultrawideband (UWB) positioning [9,10]. This technology offers a low-cost replacement to GNSS in indoor environments that can reach submeter accuracy in well designed scenarios. An example of the application of UWB ranging for cooperative navigation of multiple UAVs can be found in the work by the authors of [11]. In the field of visible light positioning systems, multiple light-emitting diodes (LEDs) beacons, placed at known position, are imaged with a conventional camera [12,13]. Each beacon is identified thanks to an ID that is modulated over the beacon light intensity: the encoded signal can be recovered from a single frame exploiting the rolling shutter property of certain imaging sensors [14]. However, for this to work, the light source must span over several pixels on the imaging sensor, restricting the operation to close range, compared to the camera resolution. The opposite idea consists in placing multiple small, point-wise active or passive targets on the moving platform and to track them from multiple static cameras placed at known positions. The targets form high contrast image features, possibly thanks to artificial IR or UV illumination and specific target surfaces, that are easily detected, e.g., by means of template matching algorithms [15]. This approach is well understood and widespread in 3D Motion Capture Systems (MOCAPs) [16] and many commercial implementations exist. The very high precision achieved by such systems, up to 1:15000 with respect to the volume diagonal [17], comes at the price of using several cameras in converging viewpoints and a very accurate intrinsic and extrinsic calibration. These systems require a heavily structured environment, and it is not straightforward to set those up outdoors, e.g., because the natural light dominates the artificial one.

In this work we focus on an alternative solution based on cooperative navigation to replace GNSS positioning without the need of structuring the environment. This is suited for outdoor operations where the GNSS position fix is unreliable because of occlusions or multipath effects. We consider two UAVs (see Figure 2): the first one, D1, flies outside of the GNSS denied area, e.g., higher or far away from any tall structure occluding the line of sight to the GNSS satellites. The second drone, D2,

flies in an area where the GNSS signals are not available, but in line-of-sight with respect to D1. This is common and realistic in many outdoor scenarios, for example when flying in the close proximity to tall objects occluding a large part of the sky. D2 navigates using the position solution estimated by D1 tracking a known pattern of high-power LEDs placed on D2 fuselage: the absolute position of D2 is obtained composing the absolute position and orientation of D1, determined by its internal INS/GNSS navigation filter and the relative D1-to-D2 position, computed via LED tracking. A feasibility study and the implications of such scheme on the accuracy of the reconstructed 3D models from D2 imagery were discussed in the work by the authors of [18], whereas the first prototype was presented in the work by the authors of [19]. A similar concept has been explored in [20], where multiple "father" UAVs help to localize a "son" UAV, fusing opportunistic GNSS observations with collaborative measurements. With respect to this work, we present and evaluate a real-world implementation of a visual ranging algorithm that allows to reduce the requirements to only one "father" UAV.

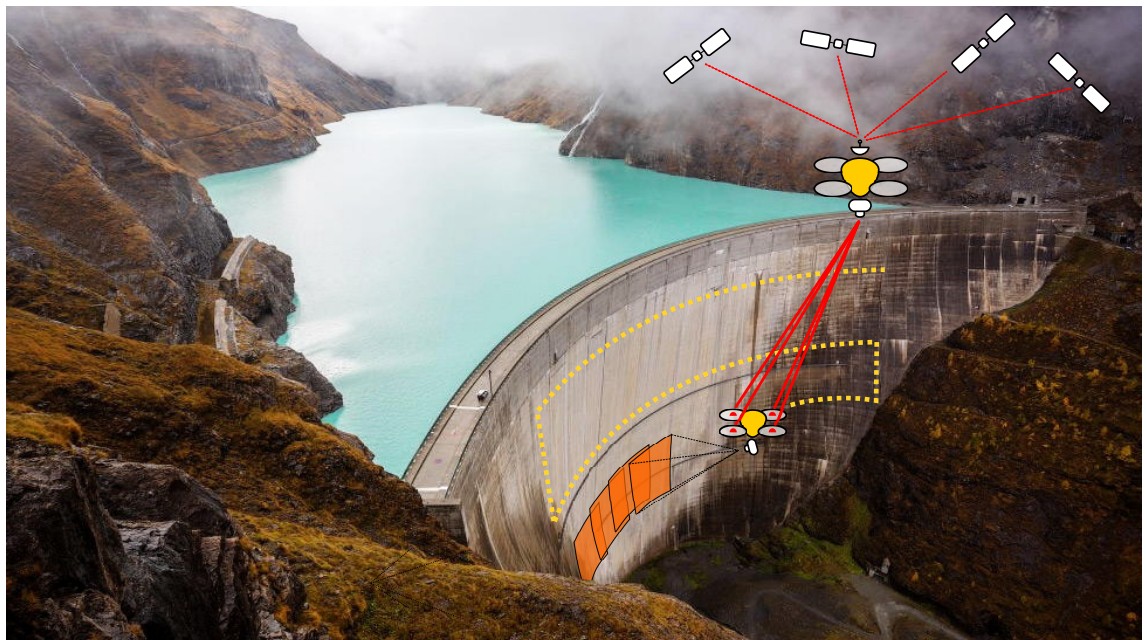

**Figure 2.** Depiction of the proposed aerial tandem performing a close-up inspection of a dam, where GNSS position fix is likely to be not available or unreliable because of sky occlusion and multipath.

As in many other collaborative navigation systems [21–23], the key element is how the robots gain knowledge of their relative position in real-time. Many methods have been explored, from laser ranging to UWB, whereas the most widely employed one consists in optical targets. Planar coded targets [24–26] generally consist in a high contrast geometric feature, such as a square or a circle, in which a code is embedded to exclude false matches and to distinguish between multiple targets present in the scene. The a-priori notion of the physical dimensions of the target, along with the intrinsic camera calibration, allows to determine the relative pose of the target with respect to the camera using a single image. One well known method is to locate known points on the target and then solve the Perspective-n-Points (PnP) problem [27] using the established set of 2D to 3D correspondences. Planar targets are widely employed in computer vision and robotics, as 3D landmarks, for camera calibration, and in several other applications in which the environment can be structured to ease tasks such as, for instance, navigation and object manipulation. One drawback is that typically many pixels are needed to recover the target code, limiting to close-to-medium distance applications. Second, the conventional planar targets are typically printed on rigid surfaces which are not applicable to flying drones because of dimensions and aerodynamic limitations.

## 2. Contributions

In this work, we present the computer-vision pipeline necessary to determine the position of an UAV outdoors and in real-time, using a single camera placed on a second UAV that flies above the first (up to 100 m, with the chosen set-up of camera and lenses). In this scenario, a large enough planar optical target would not be viable because of the limitations imposed on the vehicle aerodynamics. Thus, we chose to place small yet powerful light sources, such as light-emitting diodes (LEDs) at known asymmetric locations on the UAV fuselage, see Figure 3.

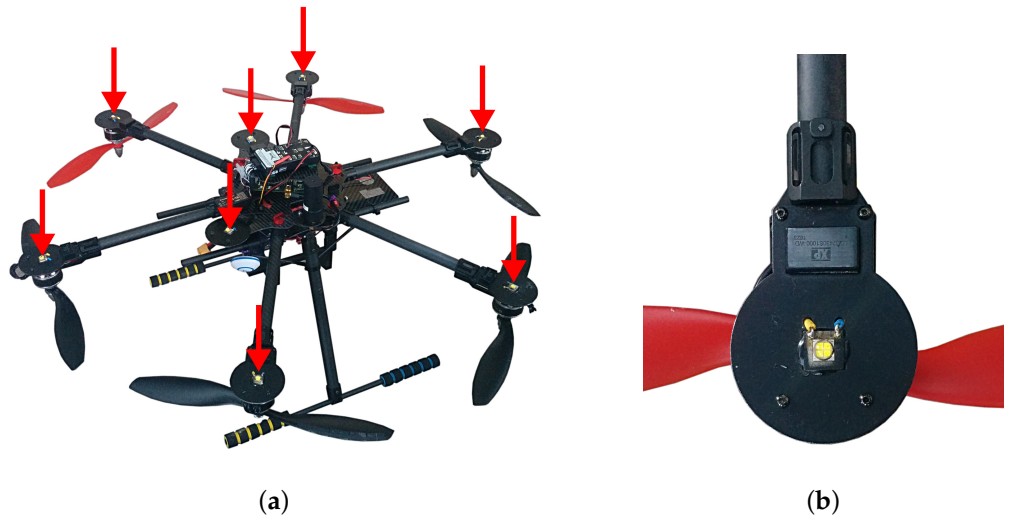

(**a**) (**b**)

**Figure 3.** Six light-emitting diodes (LEDs) are mounted on the tip of each arm of an hexacopter, plus other two on the body, in an asymmetric pattern. A 6 cm circular plate covers the background in the vicinity of the LEDs. (**a**) LED positions on the unmanned aerial vehicle (UAV). (**b**) Detail of the LED mounting.

In contrast with MOCAPs and indoor visible light positioning systems, in our application the distance is long compared to the camera resolution. Moreover, no converging viewpoints are available, so the target depth is weakly controlled and ultimately depends on how accurately the position of the light sources can be determined in the image. In our case, the image features corresponding to the LEDs consist only of a few pixels and are blurry because of motion. Thus, with classical template matching or centroid methods [28] it is difficult to achieve a precision better than one pixel.

The presented method is composed of four key innovations with respect to state-of-the-art:

1. We first derive a parametric image formation model that predicts the intensity of each pixel in the surrounding of the projection of a generic point-wise light source subject to motion blur and considering a nonideal lens system (Section 3).
2. Given that light sources are predicted to appear as bright spots in the image, with a given shape, size, and blur kernel, it is possible to use the image formation model in the target detection step: close bright pixels are grouped together and the resulting clusters are ranked according to their overall similarity to a reference image patch generated with the image formation model. Next, the a-priori notion of the shape of the LED array is introduced to remove extra false candidates and to associate each image feature to the correct point in the object space (Section 5).
3. Once the set of 2D to 3D correspondences has been established, we introduce an extension of the classical iterative algorithms for the PnP problem to determine the camera pose (i.e., its position and orientation) with respect to the LED array. Such extension relies again on the image formation model, and, instead of minimizing the reprojection error of the object points, given the camera pose, it minimizes the photometric error associated to each pixel in the vicinity of the LED projections. This allows to determine a refined, subpixel location of the object points jointly

with the camera pose, as well as the parameters of the image formation model, such as the blur kernel (Section 4).

4. Finally, we discuss a method to control the camera exposure in real time to ensure that the LEDs are always visible and with optimal brightness in the image, avoiding pixel saturation, and thus violations in the image formation model assumptions (Section 6).

In Section 7 we present an experimental evaluation of the computer vision pipeline. We show the full viability of the approach and we compare with state-of-the-art methods where possible.

## 3. Image Formation Model

In this section we derive an image formation model for a moving point-wise light source, considering a nonideal lens system. We develop the model in the one-dimensional case for simplicity, being the 2D case an intuitive extension.

At any given moment, a point-wise light source projects to a location $x$ (e.g., in pixels) on the imaging sensor. Suppose now that its apparent motion is uniform, with velocity $v$, and centered at the origin. If the lens system is ideal, the light energy at any continuous location $x$ of the imaging sensor during the exposure time $t_{\mathrm{exp}}$ is given by

$$E_{\mathrm{id}}(x) = \frac{c}{vt_{\mathrm{exp}}} \Pi \left( \frac{x}{vt_{\mathrm{exp}}} \right), \tag{1}$$

where $\Pi(\cdot)$ is the rectangular function, $t_{\mathrm{exp}}$ is the exposure time, and $c$ is a proportionality constant. Note that $\lim_{vt_{\mathrm{exp}} \to 0} \int_{-\infty}^{\infty} E_{\mathrm{id}}(x)dx = c$. The measured intensity of each pixel is typically proportional to the integral of the light energy over the pixel surface. In case of a nonideal lens system, the actual energy distribution is obtained convolving Equation (1) with the Point Spread Function (PSF) of the lens system. Here we assume that the PSF can be modeled with a Gaussian, i.e., $PSF(x) = \exp(x, \sigma) = (\sqrt{2\pi}\sigma)^{-1} \mathrm{e}^{-x^2/2\sigma^2}$ [29]:

$$E(x) = E_{\mathrm{id}}(x) * PSF(x) = \int_{-\infty}^{\infty} \frac{c}{vt_{\mathrm{exp}}} \Pi \left( \frac{x - \tau}{vt_{\mathrm{exp}}} \right) \exp(\tau, \sigma)d\tau =$$

$$= \frac{c}{vt_{\mathrm{exp}}} \left[ \int_{\infty}^{x + \frac{v}{2}t_{\mathrm{exp}}} \exp(\tau, \sigma)d\tau - \int_{\infty}^{x - \frac{v}{2}t_{\mathrm{exp}}} \exp(\tau, \sigma)d\tau \right] =$$

$$= \frac{c}{vt_{\mathrm{exp}}} \left[ \mathrm{erf} \left( \frac{x + \frac{v}{2}t_{\mathrm{exp}}}{\sqrt{2\pi}\sigma} \right) - \mathrm{erf} \left( \frac{x - \frac{v}{2}t_{\mathrm{exp}}}{\sqrt{2\pi}\sigma} \right) \right], \tag{2}$$

where we have used the fact that $\Pi(x) = \mathbb{1}(x + \frac{1}{2}) - \mathbb{1}(x - \frac{1}{2})$. hen the light source apparent velocity with respect to the camera is small, or when the exposure time is very short,

Equation (2) can be approximated with

$$E_{\mathrm{m}}(x) = c \exp(x, \sigma_{\mathrm{eq}}). \tag{3}$$

In fact, it can be shown that $E_{\mathrm{m}}(x) \to E(x)$ and $\sigma_{\mathrm{eq}} \to \sigma$ when $vt_{\mathrm{exp}} \to 0$, i.e., when the motion or the exposure time are small. In the limit case no approximation is introduced in Equation (3) in this case. When $vt_{\mathrm{exp}} \neq 0$, this does not hold. In order to quantify the impact of such approximation, we first compare $E(x)$ and $E_{\mathrm{m}}(x)$ in two cases: in Figure 4a, the effects of motion blur are very well captured by Equation (3). In the second case, see Figure 4b, we consider a faster motion and a sharper lens system. Here, the approximation becomes more coarse. A numerical study of the difference between $E(x)$ and $E_{\mathrm{m}}(x)$ as a function of $vt_{\mathrm{exp}}$ and $\sigma$ is shown in Figure 4c: it is possible to see that with a reasonably sharp lens system ($\sigma = 1$ px), the approximation introduced in Equation (3) are below 10% for motions of less than 5 pixels during an exposure time.

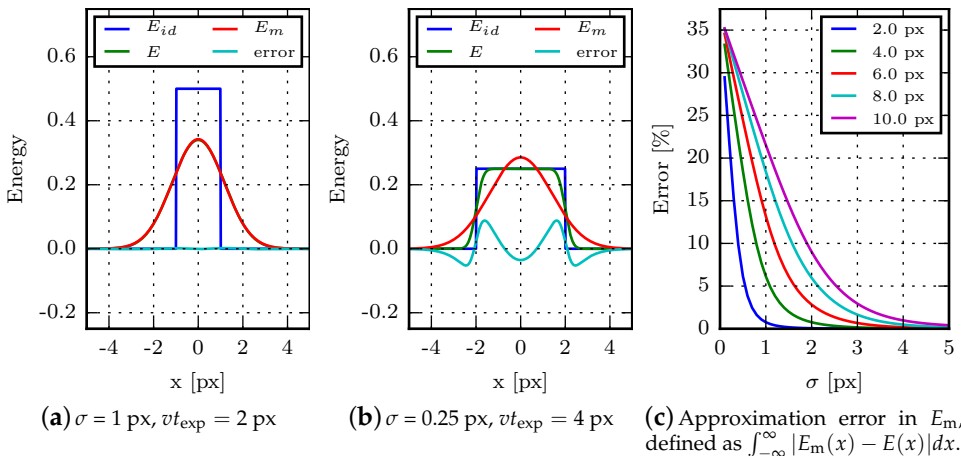

(a) $\sigma = 1$ px, $vt_{\exp} = 2$ px  (b) $\sigma = 0.25$ px, $vt_{\exp} = 4$ px  (c) Approximation error in $E_{\mathrm{m}}$, defined as $\int_{-\infty}^{\infty} |E_{\mathrm{m}}(x) - E(x)| dx$.

**Figure 4.** Comparison of $E_{\mathrm{m}}(x)$ with respect to $E(x)$ in two example cases (**a**,**b**) and numerical analysis of the difference of the two as a function of $\sigma$ and $vt_{\exp}$ (**c**).

## 4. Photometric PnP

In the following we employ the image formation model presented in the previous section to extend the classical iterative algorithms for the solution of the Perspective-*n*-Points problem [27]: if an image formation model is available for the image features associated to each known object point, the image position of such features can be determined jointly with respect to the object pose minimizing the photometric error with respect to the model. This is useful to better localize object points in the image when these consist of a limited number of pixels, and may be blurred by motion, or by a nonideal or imperfectly focused lens system.

We first review the conventional PnP algorithm. First, a set of *n* known 3D points in object space are matched with their corresponding projection in the image. In classical iterative algorithms, such as in the work by the authors of [30], or later extensions, the reprojection error is defined as the difference between the positions of the points measured in the image and the predicted ones based on the current estimate of the object pose. Such error is minimized, in least-squares sense, yielding the final object pose. More precisely, the predicted image coordinates of the *i*-th point, $[\hat{x}_i, \hat{y}_i]$ are given by the pinhole camera model:

$$\rho \begin{bmatrix} \hat{x}_i \\ \hat{y}_i \\ 1 \end{bmatrix} = K \, \Gamma_L^C \begin{bmatrix} X_i \\ Y_i \\ Z_i \\ 1 \end{bmatrix}^L, \tag{4}$$

where $[X_i, Y_i, Z_i]^L$ are the object coordinates of the *i*-th point; $\Gamma_L^C = [R_L^C | L^C]$ is the pose of the object reference frame, *L*, with respect to the camera, *C*; and *K* is the $3 \times 3$ intrinsic camera calibration matrix. As usual, $\rho$ is obtained from the third component of Equation (4) and eliminated. Lens distortion can be corrected with the well known Brown's model [31], which is omitted for brevity. Given the corresponding image observations for each point *i*, $[z_{x,i}, z_{y,i}]$, and their a priori uncertainty, $\Sigma_i$, the object pose can be determined solving

$$\hat{\Gamma}_{L,\mathrm{PnP}}^C = \arg \min_{\Gamma_L^C} \sum_i \begin{bmatrix} \hat{x}_i - z_{x,i} \\ \hat{y}_i - z_{y,i} \end{bmatrix}^T \Sigma_i^{-1} \begin{bmatrix} \hat{x}_i - z_{x,i} \\ \hat{y}_i - z_{y,i} \end{bmatrix}. \tag{5}$$

The nonlinear optimization problem in Equation (5) is typically solved by means of the Levenberg–Marquardt (LM) algorithm. An initial guess for the camera pose can be obtained by means of the direct linear transform [32,33]. Particular care needs to be taken in handling $\Gamma_L^C$ during the optimization, as the rotation component, $R_L^C$, belongs to $SO(3)$, the special orthonormal group, or the

group of rotations in three dimensions, a non-Euclidean space. $R_L^C$ is typically over-parameterized (e.g., 4D in case of unit quaternions or 9D in case of rotation matrices) and the constraints existing within the parameterization must be preserved during optimization. This problem is however well understood and manifold-aware variations of LM have been proposed, e.g., see the work by the authors of [34].

The input image coordinates, $[z_{x,i}, z_{y,i}]$, are measured separately for each point $i$ in a prior image processing step, for example by feature or template matching, corner detection, etc. The problem in this approach is that the accuracy of such image measurements is hardly below one pixel, especially in presence of motion blur and/or when the image features related to the object points consist in just a few pixels (e.g., because the distance is large compared to the camera resolution). In such cases it is also difficult to establish a consistent a priori $\Sigma_i$.

In this work we propose to determine the image coordinates of the object points jointly with the pose of $L$. Instead of minimizing the reprojection error, as defined in Equation (5), we minimize the photometric error with respect to an image formation model. In our case, each known point on the target is signalized with a small yet powerful light source. This allows us to use the image formation model derived in Section 3. We consider a small square patch of $2M + 1$ pixels centered at an (integer) initial guess of the $i$-th light source position, $[x_{i,0}, y_{i,0}]$. The intensity of the pixel $[x, y]$, with $[x, y] \in \{x_{i,0} - M, \ldots, x_{i,0} + M\} \times \{y_{i,0} - M, \ldots, y_{i,0} + M\}$ is given by

$$\hat{I}\left([x,y],[x_i,y_i]\right) = a \exp\left\{-\begin{bmatrix} x - x_i \\ y - y_i \end{bmatrix}^T \Omega \begin{bmatrix} x - x_i \\ y - y_i \end{bmatrix}\right\} + b, \tag{6}$$

where $[x_i, y_i]$ are the unknown image coordinates of the $i$-th point-wise light source. Equation (6) is the straightforward extension of Equation (3) to the two-dimensional case: $a$ is an unknown proportionality constant, $b$ is the background intensity, and $\Omega$ is a positive definite matrix encoding the blur kernel. We assume that the target is small in the image space, the background can be assumed as uniform, and that all the light sources have similar intensity. This means that $a$ and $b$ are unknown but common to each object point. The same holds for $\Omega$, provided that no fast rotation occurs around the target center viewing ray.

The target pose is determined along with the image coordinates of the points and the unknown parameters of the image formation model as

$$\hat{\Gamma}_L^C = \underset{\Gamma_L^C, [x_i, y_i], a, b, \Omega}{\arg\min} \underbrace{\sum_i \sum_{x,y} \left(zI([x,y]) - \hat{I}([x,y],[x_i,y_i])\right)^2}_{\text{Photometric error}} + \underbrace{\sum_i \begin{bmatrix} \hat{x}_i - x_i \\ \hat{y}_i - y_i \end{bmatrix}^T \Sigma_i^{-1} \begin{bmatrix} \hat{x}_i - x_i \\ \hat{y}_i - y_i \end{bmatrix}}_{\text{Geometric error}}, \tag{7}$$

where $[\hat{x}_i, \hat{y}_i]$ and $\hat{I}$ are as defined in Equations (4) and (6) and $zI$ is the pixel intensity as measured in the image. The geometric error component is equivalent to Equation (5): it is defined in terms of the unknown image positions of the points (and not with respect to image measurements) and constrains them to be consistent with the three-dimensional structure of the light source array. Indeed, $\Sigma_i$ does no longer correspond to the a priori uncertainty of the image measurements: it acts as a weight between the geometric and the photometric error components and accounts for uncertainties in the 3D positions of the object points and for the imaging system not perfectly satisfying the pinhole camera model.

The details of the solution of the optimization problem in Equation (7) are analogous to the ones of conventional PnP. However, further parameters have to be initialized: initial guesses for $a$ and $b$ can be obtained averaging the central and the corner pixels of each image patch, whereas $\Sigma$ can be set to be the identity matrix. In continuous operations, the values obtained for the last processed frames are used.

In Figure 5 we depict the results based on two real images, with the accent on the shape of the determined blur kernel; a comparison of the predicted versus actual image intensities are shown in Figure 6.

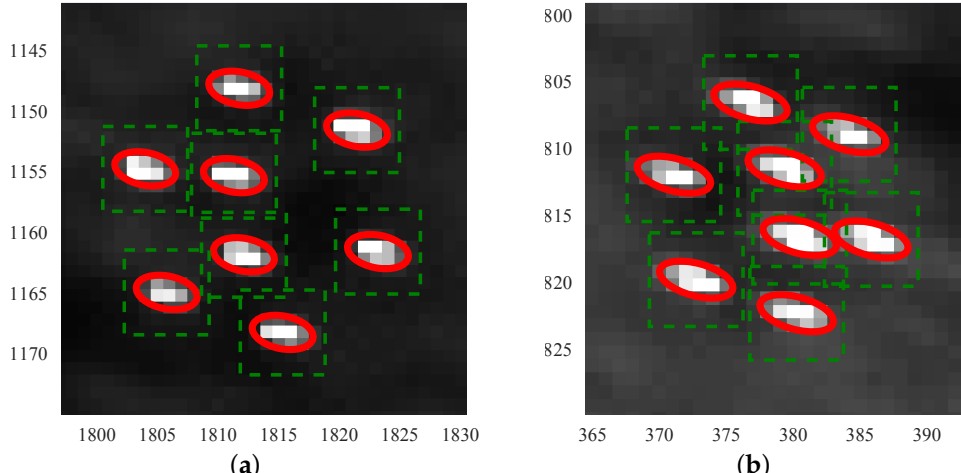

**Figure 5.** Results of fitting the image formation model on two real images of the light-source arrays; isolines at $3\sigma$ are plotted in red and image patches in dashed green. (**a**) Results at 80 m. (**b**) Results at 100 m.

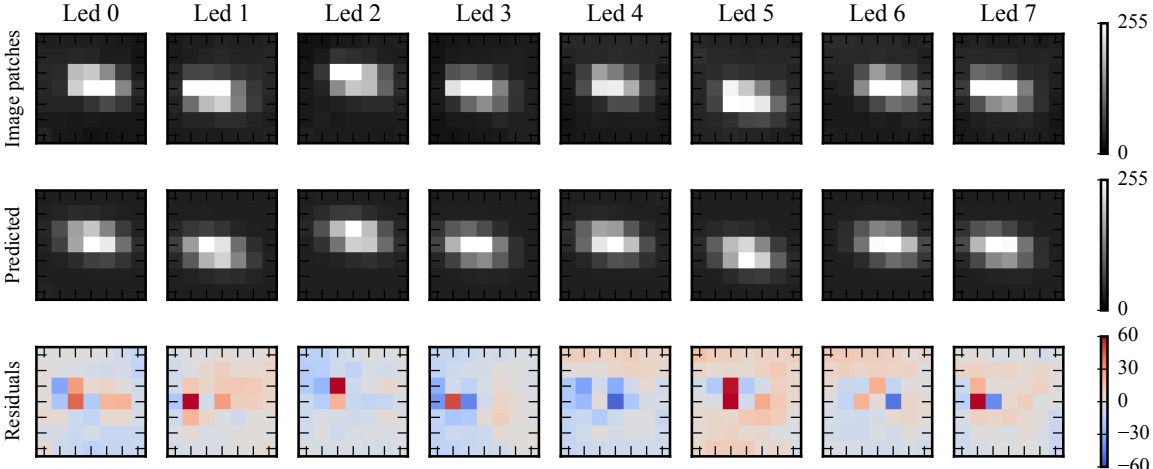

**Figure 6.** Observed versus predicted pixel intensities for the image in Figure 5a.

## 5. Target Detection

As an additional input for the solution of the Photometric PnP problem, we need to first locate the correct image features in the image, and next to establish the association of each image feature to the corresponding object point. In this work we solve this nontrivial segmentation task relying on specific aspects of the application at hand: the object points are signalized with point-wise light sources and are imaged from medium to large distance with a monochrome camera. Moreover, the tilt between the target and the camera is small, as both UAVs are hovering or gently moving in formation flight. Our solution first generates a set of promising image locations, then reduces this set based on the predicted appearance of a point-wise light source in the image, and finally establishes the association between image features and object points relying on the a priori knowledge of the geometry of the target. The details are given in the following.

Since the point-wise light sources appear as bright spots in the image, provided that a proper exposure time is set (see Section 6), the first step is to threshold the image and to cluster connected

bright pixels together. However, further objects in the scene may be bright in the image, such as highly reflective or white surfaces. To eliminate most of these candidates, we rely on the photometric model developed in Section 4: From the latest estimates of the photometric model parameters (or from default assignments, for the first frame) we can generate the expected appearance of a light source, in terms of a $2M + 1 \times 2M + 1$ patch, see Figure 6, middle row. Indeed, the parameters $a$ and $\Omega$ encode the shape and the size that a bright spot should have to correspond to one of the target signalizing lights. All clusters found are ranked according to the similarity with respect to the generated patch (in terms of squared intensity difference) and only the $K$ most similar ones are kept. Here, $K$ is a parameter of the algorithm that is chosen to be 5–10 times the number of object points $N$.

In the last step, we introduce the a priori knowledge on the shape of the target to exclude the remaining outliers and to establish the association between each image feature and the corresponding object point. For each couple of candidate image features, $c_i$ and $c_j$, we pretend that these correspond to object points 1 and 2. Based on this assumption, we compute the 2D transformation $[x, y] = T(c)$ (translation, rotation and scale) that maps the image coordinates of $c_i$ and $c_j$ to the object coordinates of points 1 and 2. Such transformation is unique and can be computed in closed form. Then, for each remaining image feature, $c_h$, we apply $T$ to obtain the expected object coordinates of $c_h$, $T(c_h)$. If the association $c_i \to 1$, $c_j \to 2$ is correct, we expect to find an object point approximately at coordinates $T(c_h)$, which we check iterating trough all the remaining object points. If we find an element for which the discrepancy is below a given threshold, we consider this a match. The couple $(i, j)$ that scores the highest number of matches, plus the matches themselves, give us the searched association between image features and object points. Note that we have assumed that a translation, rotation and scale is sufficient to approximately map image to object coordinates. In general, an homography would be needed instead, if the object points are coplanar. However, in the application at hand, the relative tilt between the two UAVs, and thus between the camera and the target, is small, making the perspective effect negligible.

The last step of the image segmentation algorithm performs an exhaustive search trough all the possible image feature to object point associations. Indeed, there is no apparent difference (e.g., color or shape) between the light sources signalizing object points and we can not rely on other distinctive image features to identify the location of the target with high confidence, as instead happens with most of the planar targets, e.g., a black and white surface with an encoded identifier, as in the work by the authors of [24]. Moreover, one ore more of the features corresponding to object points may fail to rank among the best $K$ ones according to the similarity to the reference image patch. The presented algorithm can tolerate up to $N - 3$ missing points, even tough the more points are missing the higher the chance of a wrong detection is. The complexity of the algorithm is $\mathcal{O}(K^3 N)$, where $K$ is the number of image features kept after the photometric test and $N$ is the number of object points. This moderately high complexity is bearable by a small embedded computer in real-time, provided that $K$ is kept relatively small. This is desirable, as a weaker photometric filter, and thus more candidate bright spots, increase the possibility of a wrong match: for instance, a set of bright stones on the ground could lay in a configuration comparable to the one of the object points on the target. This problem can be mitigated by increasing the number of object points. However, their density must be kept below $1/(2M + 1)^2$ points/pixel$^2$ in the whole range of operating distances, otherwise multiple ones may lie in the same image patch considered for the photometric PnP adjustment, violating model assumptions.

## 6. Automated Exposure Adjustment

In Equations (3) and (6) we introduced an image formation model for point-wise light sources able to account for motion blur and Gaussian lens point spread function. However, in practice, $zI(x, y) \in [I_{\min}, I_{\max}]$: saturated pixels are outliers with respect to that model, and can not be employed in the least-square adjustment. This means that the exposure time, $t_{\exp}$, has to be adapted to maintain a sufficient apparent brightness of the light sources while limiting pixel saturation. In the following,

we discuss how an optimal exposure time can be computed in real time based on the current estimate of the parameters of the image formation model.

It is well known that the light intensity per unit of surface decreases with the square of the distance from the source. Additionally, the light sources are generally directional, so that the emitted power decreases drastically as the angle of looking increases. The total light energy $E_{\text{tot}}$ reaching the sensor during the exposure time $t_{\text{exp}}$ can be modeled with

$$E_{\text{tot}} = \lambda \frac{t_{\text{exp}}}{\|L^C\|^2} \alpha(\Gamma_L^C), \tag{8}$$

where $\|L^C\|$ is the distance from the target and $\alpha \in [0, 1]$ is a function modeling the non-uniform light emission, which is thus dependant on the relative camera pose with respect to the target. $\lambda$ is an unknown proportionality coefficient depending on the light sources intensity. After the light sources array has been successfully detected and measured in an image, an estimate for $E_{\text{tot}}$ can be obtained from the final $a$, $b$, and $\Omega$, as in Equation (7):

$$\hat{E}_{\text{tot}} = \kappa \iint \hat{I}(x, y) - b \, dxdy = \kappa a \frac{2\pi}{\sqrt{|\Omega|}}, \tag{9}$$

where $\hat{I}$ is as in Equation (6) and $\kappa$ is a coefficient depending on the camera gain and quantum efficiency. Imposing $\hat{E}_{\text{tot}} = E_{\text{tot}}$ and $\kappa = 1$, (arbitrarily), we obtain

$$\lambda = \frac{2\pi a \left\|L^C\right\|^2}{t_{\text{exp}} \alpha(\Gamma_L^C) \sqrt{|\Omega|}}. \tag{10}$$

We continuously determine $\lambda$ using multiple frames with an exponential moving average filter so that local unmodeled effects are accounted for. Once $\lambda$ has been determined with sufficient accuracy, Equation (8) can be solved in $t_{\text{exp}}$, using the last known parameters to determine the exposure time needed to achieve the desired $E_{\text{tot}}$, which is chosen such that the number of saturated pixels is minimal, based on an average $|\Omega|$. Indeed, for the same $E_{\text{tot}}$, the peak intensity is a function of $\Omega$, as in case of motion the same light is spread over multiple pixels. While typically the relative camera pose changes slowly, the blur kernel $\Omega$ does not, as it is highly dependant on the angular velocity of camera and on vibrations.

Clearly, for a given $\lambda$ (which is related to the light sources intensity), there exist a distance for which the computed exposure time becomes excessive. The background features (e.g., the terrain) reflect the sunlight and the more $t_{\text{exp}}$ is increased, the more the highly reflective points in the background will look like the target light sources in the image. This can be solved, up to a certain density of candidates, by means of a clever image segmentation algorithm which takes into account the geometry of the array to exclude possible outliers. However, whatever algorithm will break above a certain distance.

## 7. Experimental Evaluation

We mark each tip of the arms of a hexacopter, plus other two points on the fuselage, with high-power LEDs, as in Figure 3. The maximum distance between two LEDs is 65 cm. We place the copter on the ground and we fly a second one, equipped with a nadir camera, above the first. The flight pattern has a "butterfly" shape (⋈), it is centered at the target position and enlarges with the elevation (which tops at 100 m). The camera has 5 MP resolution, a pixel size of 3.45 µm/px, the focal length of the lenses is 8 mm, so that 1 px ≈ 4.3 cm from 100 m distance. The flight was performed under sunny conditions.

We first evaluate the impact of the exposure adjustment algorithm: $t_{\text{exp}}$ needs to be controlled in real-time as a function of the current estimates of $\Gamma_L^C$, $\Omega$, $a$, and $b$ to achieve an optimal LED intensity in the image. In Figure 7, we show that as soon as the exposure adjustment algorithm is engaged,

$\hat{E}_{\text{tot}}$, as defined in Equation (8), is constant regardless of the camera pose: the farther we move from the light sources, the higher the exposure time has to be set. In the considered range of distances, the background is still substantially less bright than the light sources, which ensures that the threshold and clustering algorithm works properly.

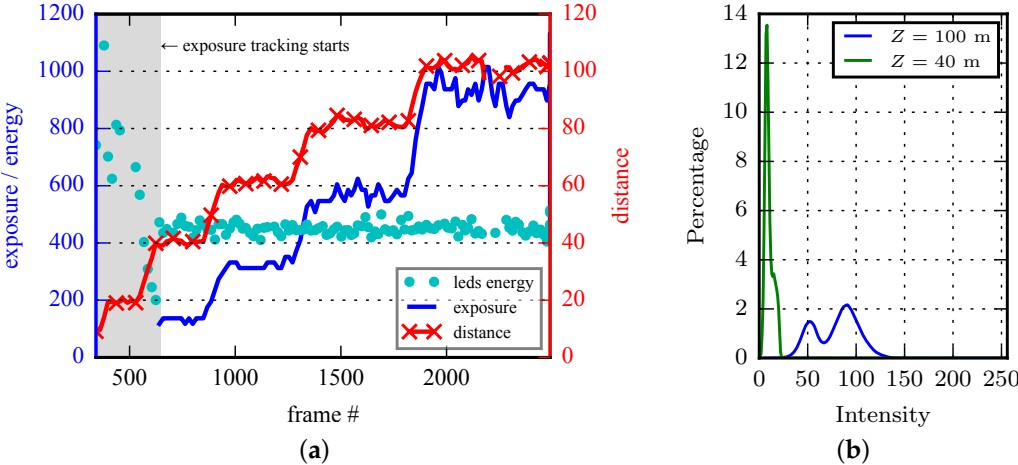

**Figure 7.** (**a**) Estimated energy $\hat{E}_{\text{tot}}$ and $t_{\text{exp}}$ compared to $\|L^C\|$. After the exposure adjustment algorithm is engaged the LEDs energy is constant regardless of the target pose. (**b**) Distribution of pixel intensities at elevation 40 m, $t_{\text{exp}} = 300$ μs and 100 m, $t_{\text{exp}} = 1000$ μs.

Next, we discuss the precision of the determined image coordinates. In Photometric PnP, $[x_i, y_i]$ are explicit unknowns in the least-squares adjustment, so we can evaluate their a posteriori uncertainty:

$$\Sigma_{\text{post}} = \frac{\nu^\top \Sigma^{-1} \nu}{R} (J^T \Sigma^{-1} J)^{-1}, \tag{11}$$

where $\nu$ is the residuals vector, $R$ is the problem redundancy, i.e., the number of observations minus the degrees of freedom, $\Sigma$ is the a priori measurement uncertainty, and $J$ is the Jacobian of the residuals with respect to the unknowns. $\Sigma$ includes the relative weight between the photometric and the geometric constraints (see again Equation (7)), and it is defined as $\Sigma = [\mathbf{1}_{2(M+1)^2}, \mathbf{0}_{2n}; \mathbf{0}_{2(M+1)^2}, k\mathbf{1}_{2n}]$, with $k = 30$, determined empirically. This means that a residual of 30 units in pixel intensity has the same weight of 1 pixel in the geometric constraint ($zI([x,y]) \in [0, 255]$). As a comparison, we run a classical iterative PnP algorithm (we used the well known implementation available in OpenCV) employing the centroid of the bright clusters as image measurements. In PnP, only $\Gamma_L^C$ is estimated, so we determine the a posteriori uncertainty of image measurements by means of covariance propagation. Note that for PnP we do not need to specify an a priori uncertainty for the image observations (which would be arbitrary), as $\Sigma = \gamma\mathbf{1}$ and $\Sigma_{\text{post}}$ does not depend on $\gamma$. The results shows that Photometric PnP allows to determine the location of the light sources with better precision compared with classical methods, $1\sigma \approx 0.05$ px, as it is shown in Figure 8.

More accurate image coordinates translate in better determination of the relative target position with respect to the camera, $\hat{L}^C$. We show this comparing the position results with respect to a classical PnP algorithm. The reference for this comparison is obtained as follows. We exploit the fact that the target is static and we orient all the images with the well know bundle adjustment software Pix4D Mapper, with the scale being fixed by camera position priors from a GPS receiver and by ground control points. Such adjustment gives the reference camera poses and the target position with respect to a global frame $W$, $\Gamma_C^W$ and $L^W$, out of which the reference $L_{\text{ref}}^C$ is computed. As hundreds of images are adjusted together, $L_{\text{ref}}^C$ is more precise than $\hat{L}^C$, which instead is determined from a single frame only. To eliminate one possible source of bias in the results, we use the same calibration in all the experiments. The result of the comparison are reported in Table 1. It is possible to see that the results are practically

unbiased, except for the $Z$ component, and that the photometric PnP algorithm outperforms classical PnP in all the statistics. Notably, we reduce the standard deviation of the error by a factor of two in the $Z$ component, which is the most sensible to the accuracy of image measurements. An equivalent comparison for the orientation estimates, $\hat{R}_L^C$, is not reported here as all planar targets suffer of pose ambiguity, meaning that in certain circumstances two orientations of the target would produce the same image projection of the known points [35], which complicates the comparison. Nevertheless, both algorithms achieve a standard deviation better than $10°$ for roll and pitch and $2°$ for yaw.

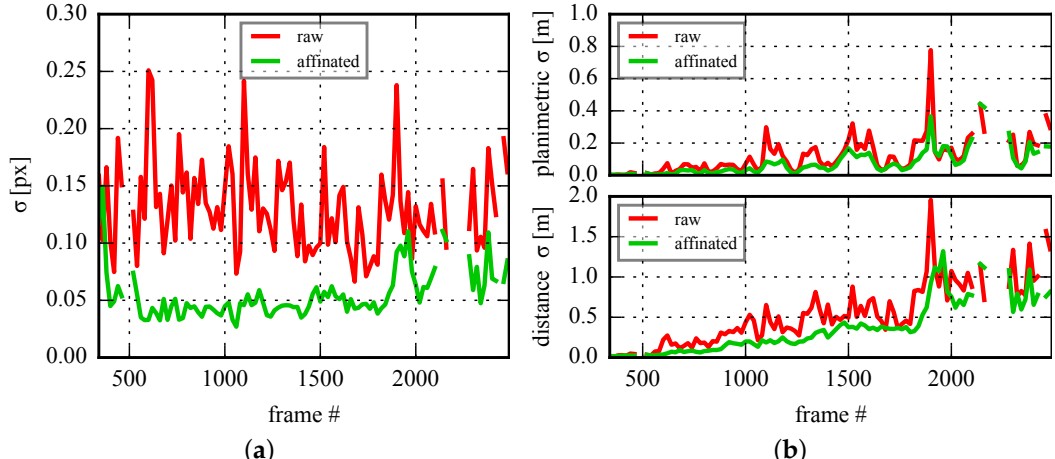

**Figure 8.** A posteriori uncertainty of the image coordinates of the light sources (**a**) and of the 3D position of the target (**b**).

**Table 1.** Relative target position error statistics (in mm).

|  | Photometric PnP | | | Classical PnP | | |
|---|---|---|---|---|---|---|
|  | *X* | *Y* | *Z* | *X* | *Y* | *Z* |
| mean | 3 | 14 | −292 | −6 | 20 | −359 |
| std | 82 | 96 | 578 | 118 | 161 | 920 |
| RMS | 82 | 97 | 647 | 118 | 162 | 986 |

## 8. Conclusions

We have presented an extension to classical algorithms for the solution perspective-n-points problem. In case an image formation model is available for the image features corresponding to the known points of the target, the photometric error can be minimized as a function of the target pose and of the unknown parameters of the model. This model was formulated for point-wise light sources subject to motion blur and nonideal lens system. We obtained an improvement of the target position precision up to a factor of two in real world experiments. As a plus, the new algorithms allows to determine the exposure time for the next frame so that the apparent intensity of the light sources is optimal for the preliminary image segmentation task, required to detect and recognize the target in the image.

**Author Contributions:** L.J. conceived and implemented the photometric pose estimation and the automatic exposure algorithms. These were implemented and tested on real prototypes thanks also to the work of A.S.-D., under the supervision of D.A.C.

**Funding:** This research received no external funding.

**Conflicts of Interest:** The authors declare no conflicts of interest.

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
