# Peer review of "Photometric Long-Range Positioning of LED Targets for Cooperative Navigation in UAVs"

_drones, doi:10.3390/drones3030069_

Round 1
Reviewer 1 Report
The paper describes a model for led target localization to achieve subpixel precision. To this aim the paper uses a target detection model that comprise photometric and geometric error.
The paper is clear and well written and propose an innovative promising technlogy for target localization. Nevertheless some issues must be addressed.
The literature about navigation techniques in challenging environment must be improved referring also to other approaches e.g.
Sundar, K., Misra, S., Rathinam, S., and Sharma, R., “Routing unmanned vehicles in GPS-denied environments,” 2017 International Conference on Unmanned Aircraft Systems (ICUAS), 2017, pp. 62–71
Y. Watanabe, A. Veillard, and C. Chanel, “Navigation and Guidance Strategy Planning for UAV Urban Operation,” in AIAA Infotech @ Aerospace, American Institute of Aeronautics and Astronautics, 2016.
F. Causa, A. R. Vetrella, G. Fasano, and D. Accardo, “Multi-UAV Formation Geometries for Cooperative Navigation in GNSS-challenging Environments,” in IEEE/ION PLANS, 2018, pp. 775–785.
V. O. Sivaneri and J. N. Gross, “UGV-to-UAV cooperative ranging for robust navigation in GNSS-challenged environments,” Aerosp. Sci. Technol., vol. 71, pp. 245–255, 2017.
Pag. 7, Equation (6) – the predicted intensity should be also dependent on the distance between the camera and the UAV, how do you model this effect?
Pag. 8, figure 6 – The figure shows the predicted and the true image of the led are not isotropic. Therefore, the relative heading between the camera and the UAV plays a role in the prediction of the pixel. Is that effect accounted in the transformation T(c) described at line 202? Do you rotate the predicted image of the leds to perform coupling with the cluster?
Pag. 8, Line 198, is K the number of led on the drone? Specify it in the paper.
Pag. 9, line 210, This line indicated the paper should report the pseudocode of the algorithm. But there isn’t any pseudocode in the manuscript.
It would be interesting to have some comparison in function of how the algorithm behave by changing the distance between the camera and the platform. There is a minimum distance needed for the algorithm to work?
Author Response
We thank the reviewer for his precious insights that helped us to improve our manuscript.
All the proposed references have been included in the introduction and put in relation with our approach.
"Pag. 7, Equation (6) – the predicted intensity should be also dependent on the distance between the camera and the UAV, how do you model this effect?"
Indeed, the predicted intensity depends on the distance from the camera. In Equation (6), this results in different values for the parameter "a", which is unknown and estimated in Equation (7). In Section 6 we suggested how "a" can be put in relation with the distance between the camera and the LED array, as correctly suggested by the reviewer, and we were able to formulate an exposure control algorithm to achieve the desired intensity of the LEDs on the image plane regardless of the camera distance.
"Pag. 8, figure 6 – The figure shows the predicted and the true image of the led are not isotropic. Therefore, the relative heading between the camera and the UAV plays a role in the prediction of the pixel. Is that effect accounted in the transformation T(c) described at line 202? Do you rotate the predicted image of the leds to perform coupling with the cluster?"
In our opinion, the apparent non-isotropy of the LED's projection on the image plane depends on the motion blur, caused by either drone motion. In the case of the lower drone, only the position counts. In the case of the upper drone, both position and orientation have an effect on motion blur. If no blur was present, as we've confirmed in static experiments, the apparent shape of the LEDs in the image plane does not depend on the drone orientation, as they are effectively point-wise light sources (the footprint of the emission is much smaller than the size of one pixel).
"Pag. 8, Line 198, is K the number of led on the drone? Specify it in the paper."
K is a parameter of the algorithm specifying how many of the ranked bright spots found in the images have to be considered further. This has been clarified in the paper.
"Pag. 9, line 210, This line indicated the paper should report the pseudocode of the algorithm. But there isn’t any pseudocode in the manuscript."
The sentence was left by mistake, we had thought that the given description of the algorithm suffices.
"It would be interesting to have some comparison in function of how the algorithm behave by changing the distance between the camera and the platform. There is a minimum distance needed for the algorithm to work?"
The experiments were performed at different distances, as shown in Figure 7. As the distance is monotonously increasing with time, Such information can be deduced from the plots in Figure 8. Indeed, the algorithm localization performances degrade as the distance increase as the very same pixel accuracy (Figure 8, left) translate in different spatial accuracy (Figure 8, right).
The algorithm cannot work if the assumption under the photometric model are violated. In particular, the main assumption is that the footprint of the light sources must correspond to less than one-pixel on the image, for those to be considered point-wise, and then the simplification in Equation (3) to hold. In our setup, this happens below 5-10 m. However, this is not a problem as it is not difficult to precisely identify the center of light-sources once these span multiple pixels in the image, e.g., with centroid algorithms.
Reviewer 2 Report
The paper is a well written article but some clarifications needed to be made before publication
It is unclear why the approximation of eqn (2) is chosen by the authors (eqn (3)), what assumptions and limitations are there must be explained.
It is really unclear what the authors mean by pose. Do they mean orientation of the target/object or the position of the object in cartesian coordinates? It is unclear how the target pose is obtained by solving the optimization problem in (7).
It would be much more convincing if the authors provided more than one test case for application of their algorithm.
Author Response
We thank the reviewer for his precious insights that helped us to improve our manuscript.
"It is unclear why the approximation of eqn (2) is chosen by the authors (eqn (3)), what assumptions and limitations are there must be explained."
Equation (2) was derived under the assumption of point-wise light source subject to an apparent constant velocity motion on the image plane. Equation (3) was obtained taking the limit of Equation (2) when the product of the point-wise light source velocity and the exposure time goes to zero. This means that Equation (3) is suited to model situations where the motion blur is limited. The impact of such approximation is discussed between lines 130 and 138 and in Figure 4.
"It is really unclear what the authors mean by pose. Do they mean orientation of the target/object or the position of the object in cartesian coordinates? It is unclear how the target pose is obtained by solving the optimization problem in (7)."
The camera ``pose'' (e.g, Gamma^L_C) is the position and the orientation of the camera frame with respect to the LED one. It is defined right after Equation (4). In Equation (7), the pose of the camera appears indirectly in \hat x_i and \hat x_i and it is (one of) the results obtained minimizing the sum of the photometric and the geometric error. One wrong reference was corrected in the text.
"It would be much more convincing if the authors provided more than one test case for application of their algorithm. "
In this work we presented detailed experiments to evaluate the performances of our algorithm in a dedicated test scenario. The reviewer may refer to
Stoven-Dubous, A.; Jospin, L.; Cucci, D.A. Cooperative Navigation for an UAV Tandem in GNSS Denied359Environments. ION GNSS (peer reviewed). The Institute of Navigation (ION), 2018.
for the discussion of real-world experiments where the lower drone was flying using only the positioning information elaborated in real-time by the algorithm presented in this work.